# Effects of mindfulness-based stress reduction on quality of life of breast cancer patient: A systematic review and meta-analysis

Xiaohui Wang[1☯], Zhicheng Dai[2☯], Xinying Zhu[1], Yu Li[1], Limin Ma[3], Xinghui Cui[4], Tongxia Zhan[1]*

1 Department of Nursing, Shandong Second Medical University, Weifang, Shandong, China, 2 Department of Clinical Medicine, Shandong Second Medical University, Weifang, Shandong, China, 3 Department of Plastic Surgery, Shandong Second Medical University, Weifang, Shandong, China, 4 Department of Nursing, Affiliated Hospital of Shandong Second Medical University, Weifang, Shandong, China

☯ These authors contributed equally to this work.
* ZTX2008@126.com

**Data Availability Statement:** All relevant data are within the manuscript and its Supporting Information files.

## Abstract

### Background

Breast cancer is the most common malignancy that occurs in women. Due to the pain caused by the disease itself and the adverse reactions in the treatment process, breast cancer patients are prone to anxiety, depression, fear of recurrence, and other negative emotions, which seriously affect the quality of life. As a systematic stress reduction therapy, mindfulness-based stress reduction is widely applied to the treatment of breast cancer patients and has been found by a growing number of studies to relieve stress, regulate mood, and improve the state. However, due to the absence of recent research and uniform outcome measures, previous studies have failed to fully explain the role of mindfulness-based stress reduction in improving the quality of life in breast cancer patients.

### Objective

We conducted a systematic review and meta-analysis to evaluate and compare the effects of mindfulness-based stress reduction therapy and standard care on the quality of life and psychological status of breast cancer patients.

### Methods

We searched PubMed, Web of Science, Embase, China's National Knowledge Infrastructure and the Cochrane Central Registry of Controlled Trials up to July 2023 to identify candidate randomized clinical trials addressing the values of mindfulness-based stress reduction in breast cancer patients.

### Results

A total of 1644 patients participated in 11 randomized controlled trials. The results of the meta-analysis showed that mindfulness-based stress reduction therapy can significantly

**Funding:** This research was supported by the Weifang Science and Technology Development Plan College Class (Grant No. 2021GX061), the Research Fund Project of the School of Nursing at Weifang Medical University (Grant No. 2022MS004), and the Weifang Medical College Nursing Research Fund 2022 Annual Project (Grant No. 2022MS003). The funders had no role in the study design, data collection and analysis, decision to publish, or preparation of the manuscript.

**Competing interests:** The authors have declared that no competing interests exist.

reduce negative emotions such as perceived stress (MD = -1.46, 95%CI = -2.53 to -0.38, $p$ = 0.03), depression (MD = -1.84, 95%CI = -3.99 to -0.30, $p$ = 0.0004), anxiety (MD = -2.81, 95%CI = -5.31 to -0.32, $p$ = 0.002), and fear of recurrence (MD = -1.27, 95%CI = -3.44 to 0.90, $p$ = 0.0004). Mindfulness-based stress reduction therapy also has certain advantages in improving the coping ability (MD = 1.26, 95%CI = -3.23 to5.76, $p$ = 0.03) and the emotional state (MD = -7.73, 95%CI = -27.34 to 11.88, $p$ = 0.0007) of patients with breast cancer.

## Conclusion

Our analyses support that, compared with standard care, mindfulness-based stress reduction therapy can significantly improve patients' coping ability, reduce adverse emotions and improve patients' emotional states.

## Introduction

Breast cancer is the most commonly diagnosed cancer type, with approximately 2.1 million new cases every year, making it the leading cause of death in women worldwide [1]. Early-stage breast cancer is considered curable [2]. At present, the most common treatment for breast cancer is surgical treatment, including breast-conserving surgery and sentinel lymph node biopsy. In addition to surgery, the main treatment methods for breast cancer include neoadjuvant chemotherapy, targeted therapy, endocrine therapy, and radiotherapy. Side effects can occur to varying degrees during treatment, such as the direct toxic effect of breast cancer treatment on the heart [3]. Other effects include changes in body and body image, arm diseases (lymphedema, axillary syndrome), and functional changes [4]. These adverse reactions can lead to a range of psychological problems in breast cancer patients, including pain, fatigue, sleep disturbances, anxiety, depression, perceived stress (PSS), cognitive dysfunction, and fear of recurrence (FOR), greatly affecting their quality of life. The evidence indicates that anxiety and depression are significantly correlated with a decreased quality of life in breast cancer patients [5].

Malignant tumor has significant effect on female mental health. Thus, female patients experience enormous mental pressure either upon learning of their malignant tumors or during the follow-up treatment [6]. One person's emotional state affects the emotional state of others, and the emotional state of cancer patients profoundly impacts the family around them and their daily activities. Negativity and depression don't just have a negative impact on the entire family [7], but also make it difficult for patients to cooperate with doctors with a positive attitude, seriously reducing their quality of life. As the most common malignant tumor with the highest incidence among women [8], breast cancer to female patients cannot be ignored. Therefore, improving the psychological status of breast cancer patients is of great significance.

As a psychologically focused therapy, mindfulness-based stress reduction (MBSR) can positively affect stress management. Patients can better manage their emotions, improve their inner focus, regulate their mental state with MBSR, and alleviating negative emotions such as loneliness, anxiety, and depression. MBSR is a widely used mindfulness-based intervention with main components such as body awareness, breathing meditation, walking meditation, mindful movement, and psychoeducation designed to relieve pain or reduce stress [9,10]. An eight week group-based therapy teaches mindfulness skills through various practices, including mindfulness of breath, thoughts, bodily sensations, sounds, and everyday activities, with

weekly 2.5-hour sessions and one full retreat day. Participants are given a CD containing instructions for home practice for 45 minutes per day, six days a week [11]. A growing body of robust evidence from several studies has demonstrated that MBSR has a significant effect on improving mental health [12,13] and is beneficial for psychological states [14–19], anxiety [20], stress [21], depression [22], and pain[23,24], particularly in high-risk pregnant women [13].

Thus, our systematic review and meta-analysis aim to comprehensively evaluate the impact of mindfulness-based stress reduction therapy on the quality of life and psychological well-being of breast cancer patients. Although the latest meta-analysis suggests a slight improvement in the quality of life through MBSR [25], it is based on data up to April 2018, omitting several relevant randomized controlled trials (RCTs) [15,26–28] and lacking information on fatigue and sleep quality. Utilizing recent RCTs, this study seeks to provide recent evidence for the clinical application of MBSR.

## Materials and methods

This systematic review and meta-analysis followed the criteria recommended by the Cochrane Collaboration and adhered to the Preferred Reporting Items for Systematic Review and Meta-Analysis (PRISMA) guidelines for reporting the results [29]. Since this study relied on published data, obtaining consent from participants was not applicable. The meta-analysis is registered with the PROSPERO database under registration number CRD42023459075.

### Literature search strategy

The population, intervention, control, and outcomes (PICO) criteria in this study were determined by the coauthors as follows: assessing the quality of life in breast cancer patients treated with MBSR compared to those receiving standard care. We conducted an analysis of both short-term and long-term quality of life in breast cancer patients. In order to identify relevant studies addressing the impact of MBSR in breast cancer patients, we conducted an integrated search of databases, including Web of Science (from 1946 to July 2023), PubMed (from 1966 to July 2023), Embase (from 1974 to July 2023), China's National Knowledge Infrastructure (from 1976 to July 2023), and the Central Cochrane Registry of Controlled Trials (from 1997 to July 2023). The search terms used were combined to capture relevant studies: (("Mindfulness"[Mesh]) OR ((("mindfulness training") OR ("Mindfulness-based Cognitive Therapy")) OR ("mindfulness-based stress reduction"))) AND ((((((breast cancer [Title/Abstract])) OR ("breast tumor")) OR ("mammary tumor")) OR ("mammary neoplasms ")) OR ("breast neoplasms")). The aim of our study was to evaluate the improvement in the quality of life in breast cancer patients treated with MBSR using a meta-analysis and to compare with those of patients receiving usual care.

### Inclusion and exclusion criteria

We assessed the quality of life using 11 outcome measures for comparison. Studies were deemed eligible if they included one of the following outcome measures: perceived pressure (PSS), depression, anxiety, fear of relapse (FOR), coping capacity, quality of life (QOL), sleep quality, post-traumatic growth (PTG), fatigue, pain, and emotional state. In the included studies, mindfulness-based stress reduction was implemented in the experimental group, while standard care was provided in the control group. We excluded meeting notes and abstracts lacking complete RCTs, as well as studies with incomplete data. In cases of replicated published studies, we selected articles with available data and the most recent results. To ensure a comprehensive search for data, we also reviewed the references of the included studies.

### Data extraction

In this study, two researchers independently screened the literature and completed data extraction. The extracted information included the first author's name, year of publication, country, sample size, intervention details, treatment duration, age, key elements of bias risk assessment, and outcome indicator data. The extracted results were then cross-checked. Any disagreements during data extraction were resolved through discussion and adjudicated by a third senior investigator. For literature lacking original data, we attempted to contact the authors to obtain the raw data; otherwise, the studies were excluded.

### Quality and risk of bias assessment

All included studies underwent quality assessment using the JADAD scale, which evaluates four aspects: random sequence generation, random hiding, blinding, and exit. The tool assigns ratings on a scale of 1 to 7, with scores of 1 to 3 considered low-quality literature and scores of 4 to 7 considered high-quality literature. To assess the risk of bias, we utilized the Cochrane Bias tool for RCTs, which includes seven assessments: sequence generation, assignment hiding, subject blindness, outcome evaluators, exit and loss of follow-up, incomplete outcome data, and selective outcome reporting. The included literature was categorized as low risk, high risk, or unclear. Two independent reviewers conducted the assessment, and any discrepancies were resolved through negotiation with a third researcher.

### Outcomes

A meta-analysis of 11 outcome indicators was performed through literature integration. Primary outcome measures included perceived stress (PSS), depression, anxiety, fear of relapse (FOR), and coping ability. Secondary outcome measures included quality of life, sleep quality, post-traumatic growth (PTG), fatigue, pain, and emotional state.

### Statistical analysis

Data were analyzed using Revman 5.3 software. Results for continuous data are presented as mean difference (MD) and 95% confidence interval (CI). Heterogeneity among studies was assessed using the $\chi^2$ and $I^2$ tests. A P value for the Q statistic of $< 0.10$ and $I^2 > 50\%$ indicated significant heterogeneity, prompting the use of a random-effects model. Subgroup analysis was conducted to explore the effects of publication date, number of participants, and duration of treatment. In cases where the data from 11 articles were expressed as a median, the algorithm of Hozo et al. [31] was employed to estimate the weighted mean and standard deviation. The test for overall effects determined statistical significance by the magnitude of the p-value, considering data as statistically significant when $p < 0.05$. Publication bias was assessed using funnel plots.

## Results

### Results of the search

The RCTs screening process for meta-analysis is illustrated in Fig 1. Initially, a systematic search yielded 1599 articles, which was reduced to 1138 after removing duplicates. Subsequently, a title and abstract filter excluded 205 non-RCTs, 16 conference abstracts, 80 meta-analysis papers, 194 irrelevant documents, 1 animal study, and 167 articles involving non-breast cancer patients. An additional 282 papers not using mindfulness-based stress reduction as an intervention were excluded. Screening the full text of the remaining 193 papers led to the

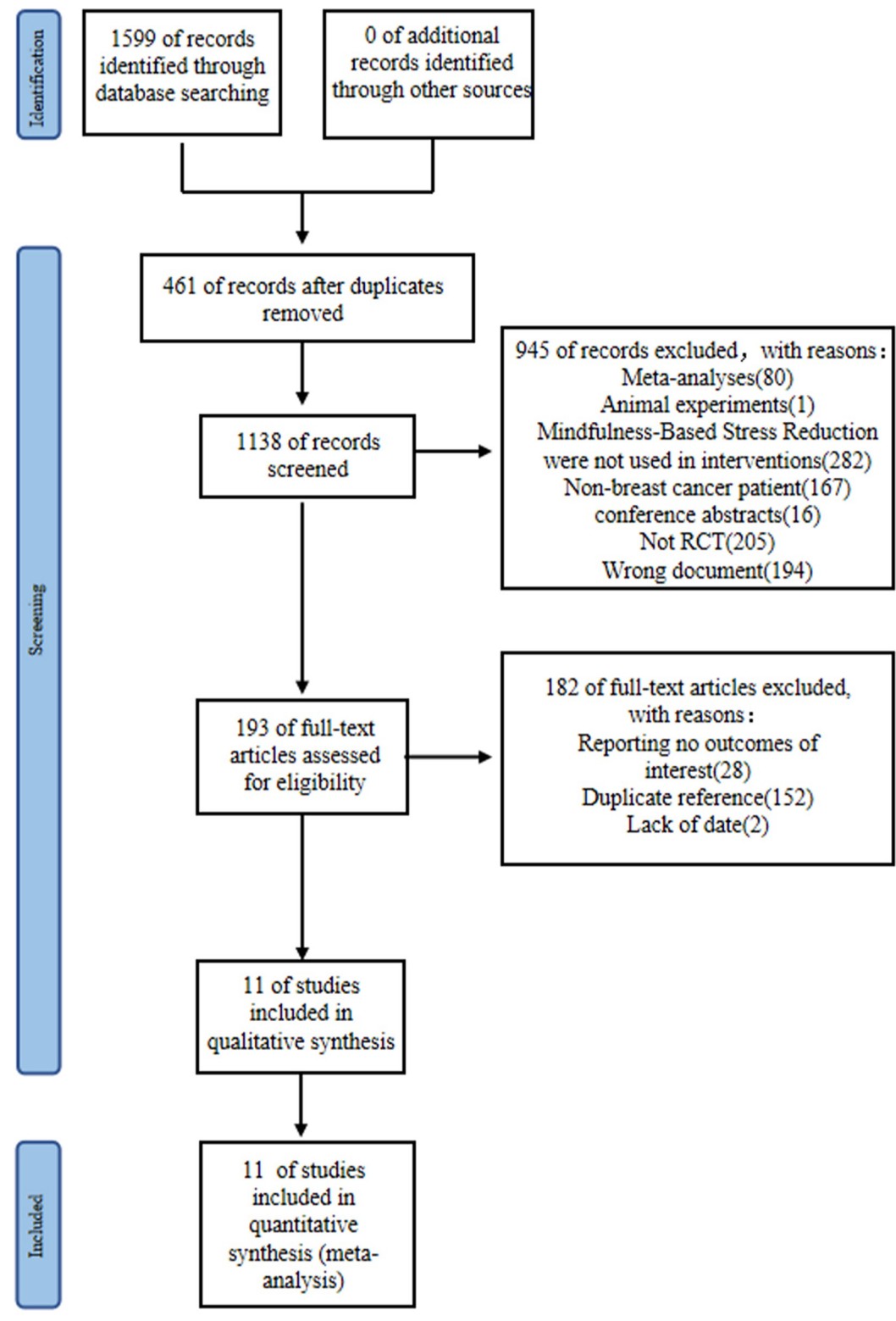

**Fig 1. Flow chart of study selection.**

exclusion of 152 duplicate references, 28 papers lacking the required outcome indicators, and 2 papers with incomplete data. Finally, 11 papers were included in this study.

## Characteristics of included studies

A total of 11 randomized controlled trials met our inclusion criteria, and detailed features of the included literature are presented in Table 1. The papers were published between 2009 [30] and 2022 [28]. Seven studies were conducted in North America countries [26,27,30–34], two in China [28,35] and two European studies [36,37]. A total of 1644 patients were included in our study, among which the study by Lengacher et al. [33] included 322 patients (the highest number of cases), while Zhang et al.[35] included 60 patients (the lowest number of cases). Four articles [30,32–34] used mindfulness-based stress reduction therapy for 6 weeks, while the remaining seven articles [26–28,31,35–37] used mindfulness-based stress reduction therapy for 8 weeks. Five articles [26,32–34,36] reported on patients with stage 0-III breast cancer, three [28,30,37] reported on patients with breast cancer during chemotherapy, one [27] reported on neuropathic pain in breast cancer patients, and the remaining two articles focused on stage I-II [31] and stage I-III [35] breast cancer patients, respectively.

## Participants and interventions

The details are given in Table 1.

## Primary outcomes

**Perceived stress.** The perceived stress (PSS) is a 10-item scale that assesses the degree to which life experiences are appraised as uncontrollable [38]. Four studies [26,30,34,35] reported the PSS (Fig 2). A meta-analysis showed that MBSR therapy significantly reduced PSS in breast cancer patients compared to standard care (MD = -1.46, 95%CI = -2.53 to -0.38, $p = 0.03$), showed high heterogeneity ($I^2 = 67\%$, $p < 0.00001$). We performed a subgroup analysis based on publication date and number of participants. Subgroup analysis (Table 2) showed that MBSR had a significant advantage over standard care in reducing PSS in breast cancer patients. Three articles [34,39,40] reported stress in patients with breast cancer after 3–6 months of follow-up (Fig 3), after 3–6 months of follow-up, MBSR showed no difference in improving PSS in breast cancer patients compared to number of participants. Subgroup analysis (Table 2) showed that MBSR had a significant advantage over standard care (MD = -1.62, 95%CI = -3.48 to 0.23, $p = 0.08$), showed high heterogeneity ($I^2 = 60\%$, $p = 0.010$).

**Depression.** Depressive symptoms were measured by the 20-item Center for Epidemiological Studies Depression Scale [41]. Three studies [26,30,33] reported the depression (Fig 2). A meta-analysis showed that MBSR therapy significantly reduced depression in breast cancer patients compared to standard care (MD = -1.84, 95%CI = -3.99 to -0.30, $p = 0.0004$), showed high heterogeneity ($I^2 = 87\%$, $p < 0.00001$). Subsequently, we performed a subgroup analysis based on publication date and number of participants. Subgroup analysis (Table 3) showed that MBSR was more effective than standard care in reducing depression in breast cancer patients. Two articles [33,39] reported depression in patients with breast cancer after 3–6 months of follow-up (Fig 3), after 3–6 months of follow-up, MBSR showed no difference in improving depression in breast cancer patients compared to the usual care (MD = -0.26, 95% CI = -1.62 to 1.09, $p = 0.93$), showed low heterogeneity ($I^2 = 0\%$, $p = 0.70$).

**Anxiety.** Anxiety measured by the State-Trait Anxiety Inventory, higher scores are indicative of more anxiety. Three studies [30,33,35] reported the anxiety (Fig 2). A meta-analysis showed that MBSR therapy significantly reduced anxiety in breast cancer patients compared to standard care (MD = -2.81, 95%CI = -5.31 to -0.32, $p = 0.002$), showed high heterogeneity

Table 1. Characteristics of the included studies.

| First Author | Country | Participants | Intervention methods | | Number of patients (Intervention/control) | Treatment duration | Age Ratio | | Follow-up time | JADAD | Outcomes |
|---|---|---|---|---|---|---|---|---|---|---|---|
| | | | Observation group | Control group | | | Observation group | Control group | | | |
| Henderson (2012) | USA | Stage I-II breast cancer | ①For 8 weeks, 7 sessions per week of 2.5 to 3.5 hours of therapy and the 6th session of 7.5 hours of intensive silent retreat therapy. ② After completing the MBSR, participants are offered three 2-hour sessions per month. | Usual care | 53/58 | 8 weeks | 49.8±8.4 | 49.8 ± 8.4 | 24 months | 4 | Coping Capacity |
| Hoffman (2012) | UK | Patients with stage 0-III breast cancer who have completed treatment for 2 months to 2 years | ①Courses lasting 8 weeks, 2 hours per week (the first and last courses are 2.25 hours). ②Week 6 plus a 6-hour mindfulness session. Do home exercises for 40 to 45 minutes six or seven days a week. | Usual care | 114/115 | 8 weeks | 49.0±9.3 | 50.1±9.1 | 3 months | 6 | Emotional States |
| Janusek (2019) | USA | Breast cancer | ①For 8 weeks (2.5 hours per week). ②6 hour meditation retreat after the fifth week. | ACC Group: Group education lasting 8 weeks, 2.5 hours per week | 63/61 | 8 weeks | 55.0 ± 10.1 | 55.2 ± 10.1 | 6 months | 7 | Sleep Quality, PSS, Depression, Fatigue |
| Lengacher (2009) | USA | Stage 0-III breast cancer patients receiving radiotherapy or chemotherapy after surgery | ①Two hour sessions six times a week for six weeks. ②Practice meditation techniques for 15–45 minutes a day. | Routine care, providing mindfulness-based stress reduction after the intervention. | 41/43 | 6 weeks | 56.1±9.1 | 58.0±10.2 | 6 weeks | 4 | FOR, Anxiety, Depression, PSS, QOL |
| Lengacher (2015) | USA | Stage 0 to III breast cancer patients | ①Two hour sessions six times a week for six weeks. ②Practice meditation techniques for 15–45 minutes a day. | Routine care, providing mindfulness-based stress reduction after the intervention. | 38/41 | 6 weeks | 56.1±9.1 | 58.0±10.2 | 3 months | 5 | Sleep Quality |
| Lengacher (2016) | USA | Stage 0 to III breast cancer patients | ①For 6 weeks, attend a 2-hour course once a week. ②Practice 15 to 45 minutes a day. | Routine care, providing mindfulness-based stress reduction after the intervention. | 155/167 | 6 weeks | 56.5±10.2 | 57.6±9.2 | 3 months | 6 | Pain, Fatigue, Depression, Anxiety, PSS, FOR |
| Reich (2017) | USA | Stage 0 to III breast cancer patients | ①A two-week course in formal meditation techniques. ②Practice 15–45 minutes a day. | Usual care | 167/155 | 6 weeks | 56.6 | 56.6 | 3 months | 4 | Depression, PSS, Sleep Quality, Anxiety, QOL, Fatigue, Pain, FOR |

(Continued)

**Table 1.** (Continued)

| First Author | Country | Participants | Intervention methods | | Number of patients (Intervention/control) | Treatment duration | Age Ratio | | Follow-up time | JADAD | Outcomes |
|---|---|---|---|---|---|---|---|---|---|---|---|
| | | | Observation group | Control group | | | Observation group | Control group | | | |
| Sarenmaim (2017) | Sweden | Breast cancer patients during chemotherapy | ①Self-directed mindfulness-based stress reduction course lasting 8 weeks.②Teachers and weekly group meetings. | Usual care | 62/52 | 8 weeks | 57.2±10.2 | 57.2±10.2 | 3 months | 7 | QOL, PTG, Coping Capacity |
| Shergill (2022) | Canada | Breast cancer survivors with nerve pain | ①Courses lasting 8 weeks, 2.5 hours per week.②One day (about 6 hours) of retreat a week. | Usual care | 49/49 | 8 weeks | 51.3±11.4 | 55.1±9.6 | 3 months | 7 | PSS, Pain, Emotional States |
| Zhang (2017) | China | Stage I-III breast cancer patients | ①For 8 weeks, 2 hours of treatment per week.②40–45 minutes of homework 6 or 7 days a week. | Usual care | 30/30 | 8 weeks | 48.67 ± 8.49 | 46.00 ± 5.12 | 3 months | 6 | PSS, Anxiety, PTG |
| Zhu (2022) | China | Breast cancer patients who have completed their first chemotherapy after surgery | Mindfulness training was performed six days a week (at least 30 minutes) for eight weeks. | Usual care | 50/51 | 8 weeks | 47.96 ± 8.51 | 49.78 ± 7.48 | / | 6 | PTG |

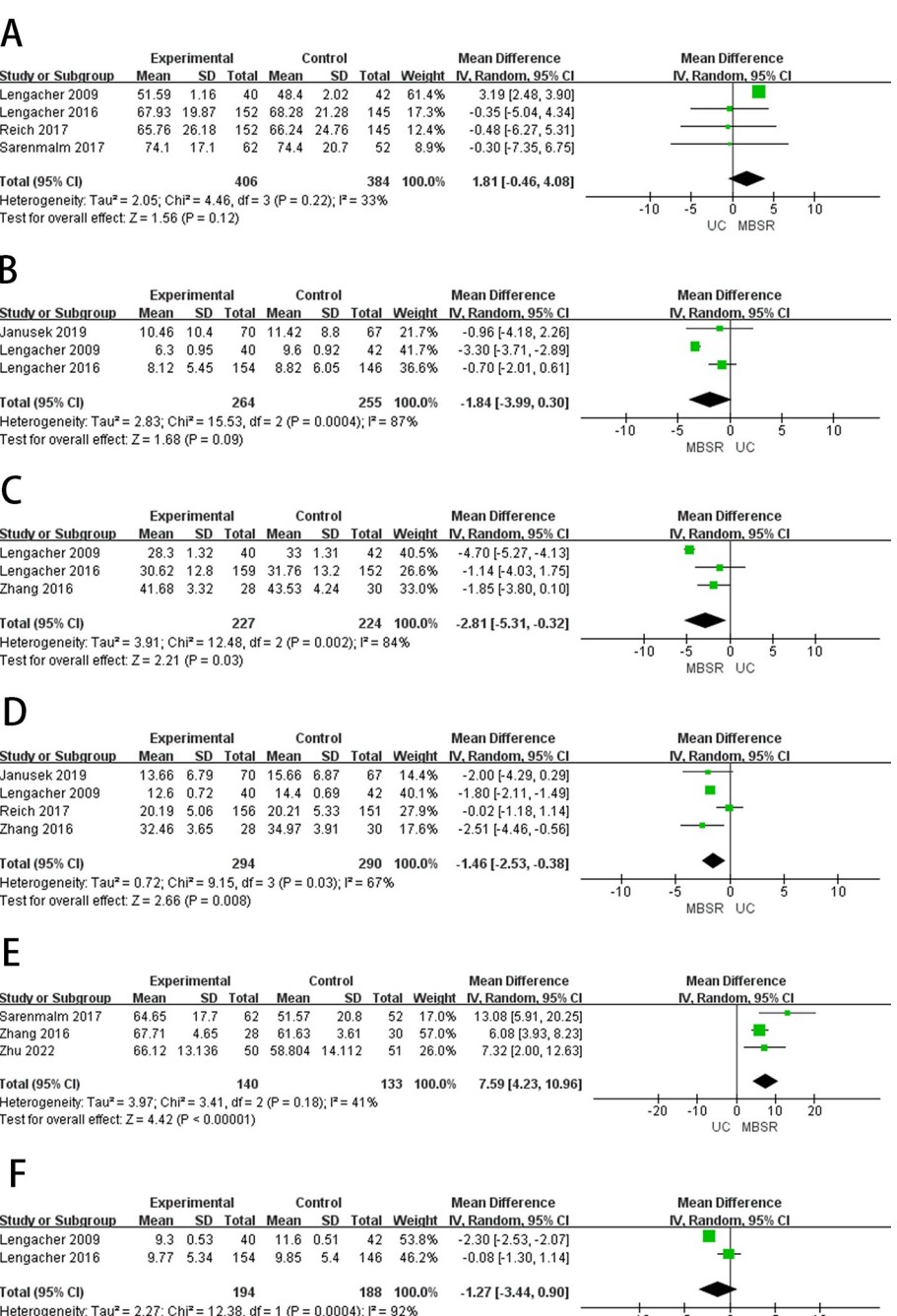

**Fig 2. The forest plots of primary outcomes.** (A) HQOL (B) Depression (C) Anxiety (D) Perceived Pressure (E) Personal Growth (F) Fear of relapse.

($I^2 = 84\%$, $p < 0.00001$). We performed a subgroup analysis on the number of participants. Subgroup analysis (Table 3) showed that MBSR was more effective than standard care in reducing anxiety in breast cancer patients. Two articles [33,40] reported anxiety in patients with breast cancer after 3 months of follow-up (Fig 3), after 3 months of follow-up, MBSR showed no difference in improving anxiety in breast cancer patients compared to standard care (MD = -2.40, 95%CI = -3.98 to -0.83, $p = 0.32$), showed low heterogeneity ($I^2 = 1\%$, $p = 0.003$).

**Table 2. Subgroup analysis of PSS.**

| Subgroups | PSS | | | | | | |
|---|---|---|---|---|---|---|---|
| | Studies, n | Participants, n | $I^2$ | Q-test | Mean difference | 95% CI | P |
| Publication date | | | | | | | |
| >2017 or = 2017 | 2 | 444 | 56 | 0.13 | -0.75 | -2.63,1.12 | 0.43 |
| <2017 | 2 | 140 | 0 | 0.48 | -1.82 | -2.12,-1.52 | <0.00001 |
| Treatment duration | | | | | | | |
| <7 weeks | 2 | 389 | 88 | 0.004 | -1.69 | -1.98,-1.39 | <0.00001 |
| >7 weeks | 2 | 195 | 0 | 0.74 | -2.30 | -3.78,-0.81 | 0.002 |

**Fig 3. The forest plot of primary outcomes after follow-up.** (A)HQOL(B) Depression (C) Anxiety (D) Perceived Pressure.

**Table 3. Subgroup analysis of depression and anxiety.**

| Subgroups | Depression | | | | | | | Anxiety | | | | | | |
|---|---|---|---|---|---|---|---|---|---|---|---|---|---|---|
| | Studies, n | Participants, n | $I^2$ | Q-test | Mean difference | 95% CI | P | Studies, n | Participants, n | $I^2$ | Q-test | Mean difference | 95% CI | P |
| Publication date | | | | | | | | | | | | | | |
| >2017 or = 2017 | 2 | 437 | 0 | 0.88 | -0.74 | -1.95,0.47 | 0.23 | | | | | | | |
| <2017 | 2 | 382 | 93 | 0.0002 | -2.08 | -4.62,0.47 | <0.00001 | | | | | | | |
| Number of participants | | | | | | | | | | | | | | |
| <100 | 2 | 219 | 50 | 0.16 | -2.7 | -4.70,-0.69 | 0.008 | 2 | 140 | 88 | 0.004 | -1.00 | -2.74,0.73 | 0.26 |
| >100 | 2 | 606 | 0 | 1 | -0.70 | -1.62,0.22 | 0.14 | 2 | 622 | 0 | 0.74 | -2.30 | -3.78,-0.81 | 0.002 |

**Fear of relapse.** Fear of relapse (FOR) measured by the 30-item Concerns about Recurrence Scale [42]. Two studies [30,33] reported the FOR (Fig 2). The results of the meta-analysis showed that MBSR therapy were significantly better than standard care in reducing FOR in breast cancer patients (MD = -1.27, 95%CI = -3.44 to 0.90, $p$ = 0.0004), showed high heterogeneity ($I^2$ = 92%, $p < 0.00001$).

**Coping capacity.** Coping capacity was evaluated using the Sense of Coherence scale (SOC) [43]. Two studies [31,37] reported the coping capacity (Fig 2). The results of the meta-analysis showed that MBSR therapy was significantly better than standard care in improving the coping capacity in breast cancer patients (MD = 1.26, 95%CI = -3.23 to5.76, $p$ = 0.03), showed high heterogeneity ($I^2$ = 78%, $p < 0.00001$).

## The secondary outcome

**Quality of life.** Quality of Life (QOL) was measured by the Medical Outcomes Studies Short-form General Health Survey [44]. Four studies [30,33,34,37] reported the QOL (Fig 4), and a meta-analysis showed that MBSR therapy did not make a significant difference in improving QOL compared to standard care (MD = 1.81, 95%CI = -0.46 to 4.08, $p$ = 0.22), showed low heterogeneity ($I^2$ = 33%, $p < 0.00001$). Subsequently, we performed a subgroup analysis based on publication date and number of participants. A subgroup analysis (Table 4) showed that MBSR was no less effective than standard care in improving QOL for breast cancer patients. Two articles [33,34] reported QOL after 3 months of follow-up (Fig 5), after three months of follow-up, MBSR did not differ in improving QOL compared to standard care (MD = -0.30, 95%CI = -4.04 to 3.44, $p$ = 0.46), showed low heterogeneity ($I^2$ = 0%, $p$ = 0.88).

**Sleep quality.** Sleep quality was measured by the Pittsburgh Sleep Quality Index [45]. Three studies [26,32,34] reported the sleep quality (Fig 4). Meta-analysis showed that there was no significant difference between MBSR therapy and standard care in improving the sleep quality of breast cancer patients (MD = -0.43, 95%CI = -0.67 to -0.19, $p$ = 0.69), showed low heterogeneity ($I^2$ = 0%, $p$ = 0.0004). Three articles [32,34,39] reported sleep quality in patients with breast cancer after 3–6 months of follow-up (Fig 5), after 3–6 months of follow-up, MBSR showed no difference in improving sleep quality in breast cancer patients compared to standard care (MD = -0.45, 95%CI = -0.68 to -0.22, $p$ = 0.39), showed low heterogeneity ($I^2$ = 0%, $p$ = 0.0001).

**Post-traumatic growth.** Post-traumatic growth (PTG) was evaluated using the Posttraumatic Growth Inventory (PTGI) [46]. Three studies [28,35,37] reported PTG (Fig 4), and

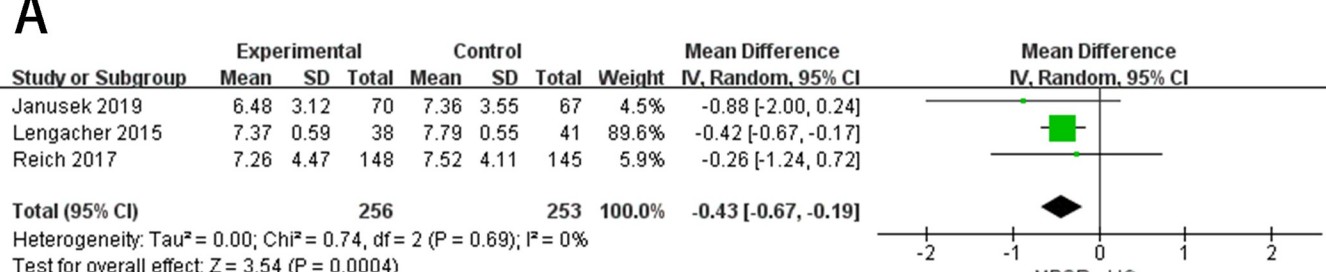

A

| Study or Subgroup | Experimental Mean | SD | Total | Control Mean | SD | Total | Weight | Mean Difference IV, Random, 95% CI |
|---|---|---|---|---|---|---|---|---|
| Janusek 2019 | 6.48 | 3.12 | 70 | 7.36 | 3.55 | 67 | 4.5% | -0.88 [-2.00, 0.24] |
| Lengacher 2015 | 7.37 | 0.59 | 38 | 7.79 | 0.55 | 41 | 89.6% | -0.42 [-0.67, -0.17] |
| Reich 2017 | 7.26 | 4.47 | 148 | 7.52 | 4.11 | 145 | 5.9% | -0.26 [-1.24, 0.72] |
| Total (95% CI) | | | 256 | | | 253 | 100.0% | -0.43 [-0.67, -0.19] |

Heterogeneity: Tau² = 0.00; Chi² = 0.74, df = 2 (P = 0.69); I² = 0%
Test for overall effect: Z = 3.54 (P = 0.0004)

B

| Study or Subgroup | Experimental Mean | SD | Total | Control Mean | SD | Total | Weight | Mean Difference IV, Random, 95% CI |
|---|---|---|---|---|---|---|---|---|
| Janusek 2019 | 9.65 | 23.54 | 70 | 10.09 | 19.98 | 67 | 5.9% | -0.44 [-7.74, 6.86] |
| Lengacher 2016 | 12.33 | 7.59 | 152 | 13.38 | 8.45 | 145 | 94.1% | -1.05 [-2.88, 0.78] |
| Total (95% CI) | | | 222 | | | 212 | 100.0% | -1.01 [-2.79, 0.76] |

Heterogeneity: Tau² = 0.00; Chi² = 0.03, df = 1 (P = 0.87); I² = 0%
Test for overall effect: Z = 1.12 (P = 0.26)

C

| Study or Subgroup | Experimental Mean | SD | Total | Control Mean | SD | Total | Weight | Mean Difference IV, Random, 95% CI |
|---|---|---|---|---|---|---|---|---|
| Lengacher 2016 | 9.59 | 9.44 | 152 | 8.28 | 8.16 | 151 | 29.6% | 1.31 [-0.68, 3.30] |
| Shergill 2022 | 3.49 | 2.09 | 49 | 3.58 | 2.61 | 49 | 70.4% | -0.09 [-1.03, 0.85] |
| Total (95% CI) | | | 201 | | | 200 | 100.0% | 0.32 [-0.93, 1.58] |

Heterogeneity: Tau² = 0.35; Chi² = 1.56, df = 1 (P = 0.21); I² = 36%
Test for overall effect: Z = 0.51 (P = 0.61)

D

| Study or Subgroup | Experimental Mean | SD | Total | Control Mean | SD | Total | Weight | Mean Difference IV, Random, 95% CI |
|---|---|---|---|---|---|---|---|---|
| Henderson 2012 | 46.8 | 1 | 53 | 43.7 | 1 | 58 | 60.9% | 3.10 [2.73, 3.47] |
| Sarenmalm 2017 | 67.7 | 12 | 62 | 69.3 | 11.5 | 52 | 39.1% | -1.60 [-5.92, 2.72] |
| Total (95% CI) | | | 115 | | | 110 | 100.0% | 1.26 [-3.23, 5.76] |

Heterogeneity: Tau² = 8.59; Chi² = 4.51, df = 1 (P = 0.03); I² = 78%
Test for overall effect: Z = 0.55 (P = 0.58)

E

| Study or Subgroup | Experimental Mean | SD | Total | Control Mean | SD | Total | Weight | Mean Difference IV, Random, 95% CI |
|---|---|---|---|---|---|---|---|---|
| Hoffman 2012 | 30.02 | 31.6 | 103 | 48.08 | 39.89 | 111 | 48.4% | -18.06 [-27.67, -8.45] |
| Shergill 2022 | 62.88 | 16.67 | 49 | 60.92 | 16.46 | 49 | 51.6% | 1.96 [-4.60, 8.52] |
| Total (95% CI) | | | 152 | | | 160 | 100.0% | -7.73 [-27.34, 11.88] |

Heterogeneity: Tau² = 182.78; Chi² = 11.38, df = 1 (P = 0.0007); I² = 91%
Test for overall effect: Z = 0.77 (P = 0.44)

**Fig 4. The forest plots of secondary outcomes.** (A) Sleep Quality (B) Fatigue (C) Pain (D) Coping Capacity (E) Emotional State.

**Table 4. Subgroup analysis of QOL.**

| Subgroups | Studies, n | Participants, n | $I^2$ | Q-test | Mean difference | 95% CI | P |
|---|---|---|---|---|---|---|---|
| Publication date | | | | | | | |
| >2017 or = 2017 | 2 | 411 | 0 | 0.97 | -0.41 | -4.88,4.07 | 0.86 |
| <2017 | 2 | 379 | 53 | 0.14 | 2.21 | 2.41,3.81 | 0.16 |
| Number of participants | | | | | | | |
| <100 | 2 | 196 | 0 | 0.33 | 3.16 | 2.45,3.86 | <0.00001 |
| >100 | 2 | 594 | 0 | 0.97 | -0.40 | -4.05,3.24 | 0.83 |

meta-analysis results showed that no significant difference could be observed between MBSR therapy and standard care in PTG in breast cancer patients (MD = 7.59, 95%CI = 4.23 to 10.96, $p$ = 0.18) and the heterogeneity was low ($I^2$ = 41%, $p$ < 0.00001).

**Fatigue.** Fatigue was measured using the Fatigue Symptom Inventory [47]. Two studies [26,33] reported fatigue (Fig 4), and meta-analysis results showed that no significant difference could be observed between MBSR therapy and standard care in reducing fatigue in breast cancer patients (MD = -1.01, 95%CI = -2.79 to 0.76, $p$ = 0.87). Showed low heterogeneity ($I^2$ = 0%, $p$ = 0.26). Two articles [33,39] reported fatigue in patients with breast cancer after 3–6 months of follow-up (Fig 5), after 3–6 months of follow-up, MBSR showed no difference in improving fatigue in breast cancer patients compared to standard care (MD = -1.12, 95%CI = -3.03 to 0.79, $p$ = 0.83), showed low heterogeneity ($I^2$ = 0%, $p$ = 0.25).

**Pain.** Pain was measured by the Brief Pain Inventory, which examines pain intensity and interference [48]. Two studies [27,33] reported the ache (Fig 4), and meta-analysis results showed that no significant difference could be observed between MBSR therapy and standard care in reducing pain in breast cancer patients (MD = 0.32, 95%CI = -0.93 to 1.58, $p$ = 0.21), showed low heterogeneity ($I^2$ = 36%, $p$ = 0.70).

**Fig 5. The forest plots of secondary outcomes after follow-up.** (A) Sleep Quality (B) Fatigue.

**Emotional states.** Emotional states measured by using the Profile of Mood State (POMS). Two studies [27,36] reported the emotional states (Fig 4). The results of the meta-analysis showed that MBSR therapy was significantly better than standard care in improving the emotional states in breast cancer patients (MD = -7.73, 95%CI = -27.34 to 11.88, $p$ = 0.0007), showed high heterogeneity ($I^2$ = 91%, $p$ = 0.11).

## Assessment of risk of bias

S1 Fig is the quality evaluation chart of the included papers. The purpose of funnel plots is to illustrate whether the included studies exhibit significant publication bias (S2–S5 Figs). One article [36] did not provide a specific description of the double-blind method, so its type was categorized as high-risk. Four articles [30–32,34] did not describe the method of concealment of allocation, but the remaining articles used characteristics such as hidden opaque envelopes or bottles for allocation. Four papers [30,31,34,35] did not detail the randomization method and were assessed as unclear. Additionally, one article [35] reported incomplete outcome data and was rated as high risk.

## Discussion

### Main results and connection with other studies

A meta-analysis of 11 outcome measures, including stress, depression, anxiety, coping ability and fear of relapse, quality of life, sleep quality, post-traumatic growth, fatigue, pain, and emotional state, was conducted through literature integration.

The results of this study revealed that MBSR therapy showed no significant difference in both short- and long-term QOL for breast cancer patients compared to usual care. This aligns with the findings of a meta-analysis by Zhang et al. [49]. In the meta-analysis of Haller et al. [50], MBSR therapy demonstrated a short-term positive effect on improving the QOL of breast cancer patients. The divergence in results could be attributed to the inclusion of a broader range of quality of life scoring tools in the Haller et al. study, while our study used a single assessment tool with eight dimensions. Not every dimension may necessarily be meaningful, contributing to the observed heterogeneity. Sarenmalm et al. [37] discovered that MBSR can enhance the short-term QOL of breast cancer patients. This difference may be attributed to the intervention duration in the study, which was 8 weeks, compared to the 6-week duration of MBSR in Lengacher et al. [30] and Reich et al. [51]. It suggests that only long-term MBSR therapy may have a positive impact on the short-term QOL of breast cancer patients.

An increasing number of doctors and nurses are turning to mindfulness-based stress reduction therapy to alleviate the negative emotions of breast cancer patients. The findings suggest that MBSR therapy effectively improves anxiety, depression, and emotional state in breast cancer patients after intervention, aligning with the findings of Hoffman et al.[52]. This positive impact on emotions is attributed to the core element of MBSR intervention, which involves meditation—a mind-based practice designed to achieve a balanced mental state [53]. Furthermore, this positive impact on emotions is attributed to the core element of MBSR intervention, which involves meditation—a mind-based practice designed to achieve a balanced mental state [50]. This implies that while MBSR therapy can offer short-term relief for negative emotions in breast cancer patients, its long-term efficacy is less apparent. A potential explanation is that the initial improvement in the status of breast cancer patients after MBSR intervention diminishes over time, overshadowed by the gradual increase in adverse reactions during treatment. Regarding promoting post-traumatic growth in breast cancer patients, MBSR shows no significant difference compared to usual care, but there is a trend toward improvement. The analysis suggests that the limited number of included studies and insufficient sample size may contribute to this observed difference. The results indicate that MBSR does not significantly

enhance short- and long-term sleep quality or reduce distress in breast cancer patients compared to standard care, consistent with the meta-analysis results by Zhang et al. [49]. The study of Janusek et al. [26] found that MBSR intervention could improve the sleep quality of breast cancer patients in the short term after 8 weeks, whereas interventions of 6 weeks by Lengacher et al. [54] and Reich et al. [51] showed less significant effects. This suggests that insufficient intervention time might impact the improvement of sleep quality in breast cancer patients. Furthermore, the study results indicate no significant difference in reducing fatigue and pain in breast cancer patients with MBSR compared standard care, potentially influenced by different disease stages. During chemotherapy, breast cancer patients may experience aggravated tumor-related symptoms, new symptoms, and an increased likelihood of psychiatric symptoms. On a positive note, the study demonstrates that MBSR significantly enhances coping skills in breast cancer patients, measured through the Sense of Coherence (SOC). Sense of Coherence reflects an individual's general perception and feeling of the outside world, representing a stable and lasting self-confidence. Higher SOC correlates with an improved ability to utilize resources and cope with pressure. Patients with higher SOC reported fewer mental symptoms, such as anxiety and depression, with a significant reduction in the prevalence of depression. The analysis attributes this improvement to MBSR techniques, such as meditation and yoga, which enhance concentration and coping skills, redirecting patients toward a healthier self-awareness and self-relationship.

MBSR demonstrated a significant reduction in negative emotions, including anxiety, depression, stress, and fear of relapse in breast cancer patients. However, there were no significant differences compared to usual care in terms of improving quality of life, sleep quality, post-traumatic growth, pain, and fatigue. All outcome indicators in this study were measured using a uniform tool, contributing to a limited number of included studies. Moving forward, it is essential to conduct large-sample randomized controlled trials, establishing a standardized evaluation method for outcome indicators. This will contribute to evidence-based insights and a clearer understanding of the impact of MBSR on breast cancer patients.

## Clinical implications

Previous studies point to MBSR significantly reducing negative emotions such as anxiety, depression, stress, and fear in breast cancer patients. Also, MBSR enhances patients' coping abilities and positively influences their emotional states. Moreover, MBSR is characterized by its low threshold for use, simplicity in implementation, and low economic cost, making it suitable for post-disease psychological intervention.Applying MBSR can contribute to a more comprehensive and improved treatment system for breast cancer patients.

## Limitations

There are limitations to our paper. The main limitation was the heterogeneity of patient populations in terms of inclusion criteria and clinical features. Breast cancer patients may be at different stages of their illness, which could be an important confounder. Different conditions and prior therapies could also influence the results. For some patients in the early stage of the disease, surgical treatment can resolve the disease, while patients in the advanced stage of the disease may be followed by targeted therapy or immunotherapy. No study was precise enough to assess whether treatment measures affect the effectiveness of MBSR at different stages of treatment. In addition, the number of outcome indicators is inconsistent between different studies, and some studies lack sufficient indicators. This variability could lead to incomplete data interpretation. As there was measurable significant heterogeneity across the studies, our conclusions should still be interpreted with caution.

## Conclusion

In conclusion, MBSR emerges as an effective intervention in significantly reducing negative emotions, including anxiety, depression, stress, and fear of relapse in breast cancer patients. Our study also demonstrates positive effects on enhancing patients' coping skills and emotional well-being. However, to strengthen the evidence base, further trials with extended follow-up periods and more tightly controlled conditions are warranted.

## Supporting information

**S1 Checklist. PRISMA-P (Preferred Reporting Items for Systematic review and Meta-Analysis Protocols) 2015 checklist: Recommended items to address in a systematic review protocol\*.**
(DOC)

**S1 Fig. RCT risk of bias summary for included randomized controlled trial.**
(TIF)

**S2 Fig. The funnel plots of primary outcomes.** (A) HQOL(B) Depression (C) Anxiety (D) Perceived Pressure (E) Personal Growth (F) Fear of relapse.
(TIF)

**S3 Fig. The funnel plot of primary outcomes after follow-up.** (A)HQOL(B) Depression (C) Anxiety (D) Perceived Pressure.
(TIF)

**S4 Fig. The funnel plots of secondary outcomes.** (A) Sleep Quality (B) Fatigue (C) Pain (D) Coping Capacity (E) Emotional State.
(TIF)

**S5 Fig. The funnel plots of secondary outcomes after follow-up.** (A) Sleep Quality (B) Fatigue.
(TIF)

**S1 Raw data.**
(DOCX)

## Author Contributions

**Conceptualization:** Xiaohui Wang, Zhicheng Dai.

**Data curation:** Xiaohui Wang, Zhicheng Dai.

**Formal analysis:** Xiaohui Wang, Zhicheng Dai, Xinghui Cui.

**Investigation:** Xiaohui Wang, Xinying Zhu, Yu Li.

**Methodology:** Xiaohui Wang, Zhicheng Dai, Xinying Zhu, Yu Li, Tongxia Zhan.

**Resources:** Limin Ma, Xinghui Cui.

**Software:** Zhicheng Dai.

**Supervision:** Limin Ma, Xinghui Cui, Tongxia Zhan.

**Validation:** Limin Ma, Tongxia Zhan.

**Writing – original draft:** Xiaohui Wang, Zhicheng Dai.

**Writing – review & editing:** Xiaohui Wang.

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
