## [Decision Letter · Decision Letter 0]

15 May 2024

PONE-D-23-40328Effects of mindfulness-based stress reduction on quality of life of breast cancer patient: A meta-analysisPLOS ONE

Dear Dr. Zhan,

Thank you for submitting your manuscript to PLOS ONE. After careful consideration, we feel that it has merit but does not fully meet PLOS ONE’s publication criteria as it currently stands. Therefore, we invite you to submit a revised version of the manuscript that addresses the points raised during the review process.

We look forward to receiving your revised manuscript.

Kind regards,

Gabriel G. De La Torre

Academic Editor

PLOS ONE

Reviewers' comments:

Reviewer's Responses to Questions

**Comments to the Author**

1. Is the manuscript technically sound, and do the data support the conclusions?

Reviewer #1: Yes

2. Has the statistical analysis been performed appropriately and rigorously? 

Reviewer #1: Yes

3. Have the authors made all data underlying the findings in their manuscript fully available?

Reviewer #1: Yes

4. Is the manuscript presented in an intelligible fashion and written in standard English?

Reviewer #1: Yes

5. Review Comments to the Author

Reviewer #1: The paper provides a feasible study of MBSR which examines several studies in relation to breast cancer. The discussion is well constructed as is the methodology. I have made several grammatical corrections to the paper. I have also omitted use of "sexist" language for the paper. Overall, the paper is well researched and written with sound conclusions. the authors also admit that MBSR results can be different depending on the stage of cancer. This is an important finding as it can provide a better understanding when MBSR is most suitable. Therefore, timing of MBSR therapy is important. Certainly, one size does not fit all, and the authors recognise this. The authors point to the limitations and strengths of MBSR based on previous studies. The paper is well constructed. I accept the paper for publication after minor grammatical errors and comments have been addressed.

6. PLOS authors have the option to publish the peer review history of their article (what does this mean?). If published, this will include your full peer review and any attached files.

Reviewer #1: **Yes: **Arthur Saniotis

---

## [Author Response · Author response to Decision Letter 0]

3 Jun 2024

Dear editor,

Thank you for your letter and for the reviewers’ valuable comments concerning our manuscript entitled “Effects of mindfulness-based stress reduction on quality of life of breast cancer patient: A systematic review and meta-analysis”. These comments are of great help to the revision and improvement of our article, and help to improve the quality of the article. All authors have carefully studied the comments and made revisions with the hope of publication in PLOS ONE. Additionally, We modified the format of the manuscript according to PLOS ONE's style requirements. The reference section has also been revised. It should be noted that the name of our school was changed from "Weifang Medical College" to "Shandong Second Medical University" last December, so I changed the unit in the article to "Shandong Second Medical University".

A list of our responses to specific recommendations follows. If there are any other modifications we could make, we would like very much to modify them and we really appreciate your help.

Yours sincerely,

Xiaohui Wang

Response to Reviewers:

Thank you very much for your revision and comments in the article. The following are my answers to your questions.

1. What do you mean by this? This is sexist langauge.the use of words such as delicate and sensitive are inappropriate for a scientific paper. I would omit this line.

Response: Thank you for your comment. As per your suggestion, we change “Malignant tumors have a significant psychological impact on patients, and women are often more delicate and sensitive than men” to “Malignant tumor has significant effect on female mental health”(Page 4, line 84).

2.What do you mean by this?? Emotions do not operate as pathogens, You can state that one's emotional state can influence the affective states of others.

Response: Thank you for your comment. We change “Emotions are contagious” to “One person's emotional state affects the emotional state of others”(Page 4, line 87 ).

3. Do not use metaphors such as casting a dark shadow. This is unscientific.

Response: Thank you for your valuable suggestion. We Change “Negative and depressed emotions not only cast a shadow on the entire family” to “Negativity and depression don't just have a negative impact on the entire family” (Page 4, line 90). 

4.Some language problems in the article.

Response: Thank you for your comment. We have made all modifications according to the annotations in the article.

5.You should state prior to 2018.

Response: Thank you for your help and valuable advice. We have moved the position of this paragraph to line 113 before "Although"

6.The correct method to write author and year is Henderson (2012). do this for the other authors.

Response: Thank you for your comment. We have adopted your comments and revised the method to write.

7. Do not use the word ''usual'. change this word to standard in the other occasions where it is used.

Response: Thank you for your valuable advice. We have revised all the usual care in this article to standard care.

8.Use the non-abbreviated name at the start of a sentence.

Response: Thank you for your valuable advice. We have made modifications according to your suggestion (Page 25, line 399).

---

## [Editor Report · Decision Letter 1]

21 Jun 2024

Effects of mindfulness-based stress reduction on quality of life of breast cancer patient: A systematic review and meta-analysis

PONE-D-23-40328R1

Dear Dr. Zhan,

We’re pleased to inform you that your manuscript has been judged scientifically suitable for publication and will be formally accepted for publication once it meets all outstanding technical requirements.

Kind regards,

Gabriel G. De La Torre

Academic Editor

PLOS ONE

---

## [Editor Report · Acceptance letter]

11 Jul 2024

PONE-D-23-40328R1 

PLOS ONE

Dear Dr. Zhan, 

I'm pleased to inform you that your manuscript has been deemed suitable for publication in PLOS ONE. Congratulations! Your manuscript is now being handed over to our production team.

Kind regards, 

on behalf of

Dr. Gabriel G. De La Torre 

Academic Editor

PLOS ONE